# Pediatric In-Hospital Cardiac Arrest: An Examination of Resuscitation Outcomes

**DOI:** 10.3390/medicina60111739

**Published:** 2024-10-23

**Authors:** Yakup Söğütlü, Uğur Altaş

**Affiliations:** 1Ümraniye Training and Research Hospital, Pediatric Emergency Medicine Clinic, University of Health Sciences, Istanbul 34764, Turkey; beyoglu@hotmail.com; 2Department of Pediatric Allergy and Immunology, Umraniye Training and Research Hospital, Istanbul 34764, Turkey

**Keywords:** in-hospital cardiac arrest, resuscitation outcomes, children, survival

## Abstract

*Background and Objectives*: We aimed to assess the outcomes of pediatric in-hospital cardiac arrests (IHCAs) and to identify key factors influencing survival. *Materials and Methods*: This retrospective, single-center study examined the demographic characteristics, symptoms, comorbidities, initial rhythm, duration of cardiopulmonary resuscitation (CPR), lactate levels, and outcomes of pediatric patients with IHCAs and compared these parameters between survivors and non-survivors. *Results*: A total of 43 patients were included in this study, including 21 boys (48.8%) and 22 girls (51.2%) with a median age of 36 months (range 1–203). CPR was initiated due to pulselessness in 23 patients (53.5%), respiratory arrest in 13 (30.2%), and bradycardia in 7 (16.3%). The first monitored rhythm in the emergency department was asystole in 29 patients (67.4%) and bradycardia in 14 (32.6%). Despite effective CPR, the mortality rate was 65.1% (*n* = 28). As a prognostic factor, asystole was found to be more common in non-survivors than in survivors (83.1% vs. 40%, *p* = 0.005). Additionally, lactate levels (16.6 vs. 10.6, *p* = 0.04) and CPR duration (45 vs. 15 min, *p* < 0.001) were significantly higher in non-survivors. *Conclusions*: IHCAs remain a critical concern, with varying outcomes influenced by factors such as initial rhythm, lactate levels, and CPR duration.

## 1. Introduction

Pediatric in-hospital cardiac arrest (IHCA) is a life-threatening event associated with high risks of morbidity and mortality. In the United States, the incidence of IHCA is estimated to be 15,200 cases annually, including 7100 pulseless cardiac arrests and 8100 non-pulseless events [1]. Despite improvements in pediatric healthcare, cardiac arrest remains a leading cause of death in children, particularly in those with chronic health conditions [2,3]. In children, cardiac arrests are more frequently characterized by respiratory failure, circulatory shock, or sepsis, as opposed to the cardiac etiologies that are more typical in adult cases [4]. Congenital heart disease, severe infections, trauma, and metabolic disorders are key contributors to pediatric cardiac arrest. Additionally, underlying conditions, such as cardiac arrhythmias and myocarditis, are increasingly recognized as significant factors in certain cases. Unfortunately, the outcomes following pediatric cardiac arrest are often poor, although advancements in pediatric resuscitation and post-resuscitation care have led to gradual improvements. The survival rate to hospital discharge ranges from 19% to 57%, with significant variability depending on the cause, initial rhythm, and setting of the arrest [3]. Nonetheless, many survivors of pediatric IHCAs experience long-term neurological impairments due to the lasting effects of hypoxia during the event. Hypoxia, resulting from inadequate oxygen supply to the brain during cardiac arrest, can cause significant and sometimes irreversible damage to brain tissue, leading to cognitive, motor, and developmental deficits. This underscores the critical importance of understanding the epidemiology, causes, outcomes, and prognostic factors associated with pediatric cardiac arrest. By deepening our knowledge in these areas, healthcare providers can enhance patient management, improve survival rates, and reduce the long-term impact of neurological impairments. Tailoring interventions based on these insights allows for the optimization of resuscitation techniques and the development of more effective post-arrest care strategies, which may involve targeted therapies to mitigate hypoxia-induced brain injury.

Given the profound clinical and societal implications, it is crucial to comprehensively evaluate the outcomes of pediatric cardiac arrests. Understanding the short- and long-term results of these events can help identify the key factors that influence survival, such as the timing and quality of resuscitation, underlying medical conditions, and the availability of advanced life support. Therefore, the aim of this study was to assess the outcomes of pediatric IHCAs to uncover critical determinants of survival and inform future strategies aimed at improving care and reducing mortality in this vulnerable population.

## 2. Materials and Methods

### 2.1. Study Design and Population

This retrospective, single-center observational study was conducted to investigate all pediatric patients who experienced in-hospital cardiac arrests (IHCAs) over a five-and-a-half-year period, from January 2019 to July 2024. This study aimed to identify factors associated with survival by analyzing key patient characteristics and clinical data. Patients who experienced cardiac arrest after being admitted to the hospital were included in the scope of this study. This inclusion criterion was established to ensure that the data collected would reflect IHCAs, allowing for a more controlled analysis of the factors influencing their outcomes. However, patients who were brought to the hospital already in cardiac arrest and subsequently transferred to the emergency department were excluded from this study. This decision was made due to the lack of reliable information on several key modifying factors, such as the quality of cardiopulmonary resuscitation (CPR) performed prior to hospital arrival, the duration of the resuscitation efforts, and the initial cardiac rhythm recorded. These unknowns could introduce variability and potentially confound the analysis of the outcomes.

Moreover, patients who experienced cardiac arrest because of trauma were also excluded from this study. This exclusion was necessary because traumatic arrests occur via different physiological mechanisms and may require different resuscitation protocols, which could skew this study’s findings when focusing on non-traumatic, in-hospital cardiac arrest cases. By narrowing the scope in this way, this study aimed to provide a more accurate assessment of the factors affecting survival and outcomes in patients (Figure 1).

### 2.2. Measures

Demographic variables such as age and sex, as well as clinical parameters including presenting symptoms, underlying comorbidities, and the first monitored cardiac rhythm, were meticulously documented. Additionally, data on the duration of CPR, lactate levels measured within the first 10 min following the initiation of CPR, and the final outcomes of the patients were collected and analyzed.

Furthermore, to better understand the factors influencing survival, comparisons were made between survivors and non-survivors across these variables. The measurement of lactate levels, which can serve as an indicator of tissue perfusion and oxygenation, was a key focus, given its potential prognostic value in the post-arrest setting.

The decision to terminate resuscitation (TOR) was made in accordance with established guidelines, specifically following the universal termination of resuscitation criteria, which consider factors such as the patient’s clinical status, time elapsed during CPR, and the likelihood of recovery [5,6].

Cardiopulmonary resuscitation was performed following the current recommendations of the American Heart Association (AHA), ensuring that the resuscitative efforts adhered to the highest standards of care [5,6]. Adherence to these guidelines helped maintain consistency in the CPR approach and allowed for reliable comparisons of the outcomes.

A neurological assessment was performed before discharge from the hospital, and the Pediatric Cerebral Performance Category (PCPC) was used to evaluate neurological outcomes.

### 2.3. Statistical Analysis

The database for this study was developed using the SPSS 29.0 software package (IBM Corp., Armonk, NY, USA), a widely utilized tool for statistical analysis in medical research. To ensure the robustness of the data analysis, the normality of the distribution of continuous variables was thoroughly assessed using both visual and analytical methods. Visual assessments were carried out by inspecting histograms and probability plots, allowing researchers to evaluate the shape of the data distribution and identify any deviations from normality. In addition, analytical assessments were performed using the Shapiro–Wilk test, which provided statistical measures to further validate whether the data followed a normal distribution pattern. For the presentation of quantitative data, two approaches were employed depending on the distribution of the variables. Variables that were normally distributed were expressed as means along with their corresponding standard deviations (SDs), while non-normally distributed variables were presented as medians with accompanying minimum and maximum values to reflect the full range of the data. Categorical variables, on the other hand, were summarized as frequencies or percentages.

For the comparison of categorical variables between groups, the Chi-square test was primarily used. However, when the expected frequencies in any of the contingency table cells were too low for the Chi-square test to be valid, Fisher’s exact test was applied to ensure accurate results. Continuous variables, which did not follow a normal distribution, were compared using the non-parametric Mann–Whitney U test. The results were considered statistically significant when the *p*-value was less than 0.05 (two-tailed).

### 2.4. Ethics

This study was conducted in accordance with the principles of the Declaration of Helsinki, which ensures ethical standards in medical research by prioritizing the safety, rights, and well-being of participants while promoting transparency and scientific integrity. Ethics committee approval was obtained by the Istanbul Health Sciences University Ümraniye Training and Research Hospital Clinical Research and Ethics Committee (Date: 11 July 2024, Number: 210).

## 3. Results

A total of 43 patients were included in this study. Of them, 21 (48.8%) were boys and 22 (21.2%) girls. The median age of patients was 36 (1–203) months. The most common presenting complaint prior to the cardiac arrest was shortness of breath, reported in 24 (55.8%) patients. This was followed by a clouding of consciousness, which was observed in 10 (23.2%) patients. Cyanosis, a critical indicator of oxygen deprivation, was noted in six (14%) patients, while other symptoms, such as vomiting, nausea, or fever, were present in three (7%) patients.

All the cardiac events occurred within the emergency department, underscoring the importance of this high-acuity setting for managing critical pediatric cases. In terms of transportation to the hospital, 23 (53.5%) patients were brought via ambulance while 20 (46.5%) were transported by their families. The median length of stay in the emergency department prior to the cardiac event was 120 min, with a range of 40 to 720 min. None of the patients had previously experienced cardiac arrest.

Twenty-three (53.5%) of the patients in this study had documented comorbid diseases. Among these, neurological disorders were the most prevalent comorbidity, and cardiac conditions were the second most common comorbidity. Table 1 provides detailed information on the specific comorbidities observed.

Cardiopulmonary resuscitation (CPR) was initiated due to pulselessness in 23 patients (53.5%), respiratory arrest in 13 patients (30.2%), and bradycardia in 7 patients (16.3%). The first monitored cardiac rhythm detected before CPR revealed that asystole was present in 29 patients, representing 67.4% of the cohort. In contrast, 14 patients (32.6%) exhibited bradycardia. Despite the implementation of effective CPR, the overall mortality rate was 65.1%, corresponding to 28 patients. The median duration of CPR was 45 min, with a range of 2 to 120 min. Additionally, the median lactate level recorded during resuscitation was 14.2 mmol/L, with values ranging from 1.2 to 27 mmol/L.

When comparing the various clinical and demographic variables between survivors and non-survivors, no significant differences were observed between the two groups regarding age, sex, the presence of comorbidities, or the reasons for initiating CPR. However, notable differences emerged in the first monitored cardiac rhythm, where asystole was significantly more prevalent among non-survivors, with 83.1% of non-survivors presenting with this rhythm compared to only 40% of survivors (*p* = 0.005). Additionally, lactate levels at the time of resuscitation were significantly higher in non-survivors, with a median lactate level of 16.6 mmol/L compared to 10.6 mmol/L in survivors (*p* = 0.04). Furthermore, the duration of CPR also showed a significant disparity, with non-survivors receiving CPR for a median of 45 min, while survivors had a median duration of only 15 min (*p* < 0.001) (Table 2). All survivors achieved a good neurological outcome upon discharge. Among the surviving patients, ten had a pre-existing chronic condition that affected their neurological development. No worsening of neurological findings was observed in these patients. The remaining five patients had normal neurological function as assessed by the PCPC.

## 4. Discussion

The survival outcomes of pediatric IHCAs are influenced by a multitude of interrelated factors that can significantly impact the likelihood of a favorable recovery. These factors include the patient’s underlying diagnoses. Additionally, the initial cardiac arrest rhythm plays a vital role in predicting survival. The quality of CPR administered is another critical determinant, as effective chest compressions and timely interventions can significantly enhance the chances of restoring circulation. Furthermore, the achieved hemodynamic status during and after resuscitation, including factors such as blood pressure and circulation, is essential for ensuring adequate perfusion to vital organs. The duration of CPR is also an important consideration; prolonged resuscitation efforts may indicate a more complex clinical scenario that can influence overall outcomes. Lastly, the standard of post-cardiac arrest care, including interventions aimed at preventing neurological damage and supporting recovery, can have a profound impact on long-term survival and quality of life for these patients. In our analysis, we demonstrated a significant association between survival and several key factors, notably the initial monitored rhythm, elevated lactate levels, and a prolonged duration of CPR. Specifically, the presence of certain rhythms at the outset of monitoring was correlated with better survival rates, while elevated lactate levels indicated more severe tissue hypoxia and were associated with poorer outcomes. Moreover, a prolonged duration of CPR is generally linked to unfavorable outcomes.

According to previous pediatric studies, cardiac arrests typically occur in children with a median age of 1 to 2 years and a mean age of 3 to 5 years, with a slight predominance of males [2]. In the present study, a total of 43 patients were evaluated. Of these, 21 were boys while 22 were girls, resulting in a near-equal sex distribution. The median age of the patients was 36 months, with the range spanning from 1 to 203 months. This wide age range indicates that this study encompassed not only infants and toddlers but also older children, reflecting various developmental stages.

Pediatric IHCA typically occurs in the context of progressive respiratory failure or shock. The most common initial cardiac arrest rhythms are usually non-shockable (e.g., asystole, pulseless electrical activity [PEA], or bradycardia with poor perfusion). Bradycardia with poor perfusion serves as the initial CPR rhythm in >50% of IHCA cases. Children with bradycardia and poor perfusion tend to have better survival outcomes compared with those who are pulseless at the onset of CPR [2]. According to the data from Get with The Guidelines-Resuscitation, a total of 5592 pediatric patients received CPR, with 50.1% receiving bradycardia with poor perfusion and 49.9% receiving initial pulseless cardiac arrest. Among those with bradycardia, 31.0% became pulseless after a median of 3 min of CPR. The survival rates to discharge were 70.0% for patients with bradycardia and a pulse, 30.1% for those with bradycardia progressing to pulselessness, and 37.5% for initial pulseless cardiac arrest [7]. The American Heart Association National Registry of CPR Investigators also compared 1853 (55%) patients receiving chest compressions for bradycardia/poor perfusion with 1489 (45%) patients receiving CPR for asystole/pulseless electrical activity (PEA) [8]. They concluded that CPR for bradycardia with poor perfusion was associated with a higher rate of survival to hospital discharge [8]. In the present study, when examining the reasons for CPR, respiratory arrest was more common among survivors, whereas pulselessness was more common among non-survivors. Furthermore, ventricular fibrillation (VF) and pulseless ventricular tachycardia (PVT) account for approximately 10% of all CPR events [9]. Both primary VF and PVT are associated with more favorable outcomes than PEA and asystole. In our study, although there were no cases of VF or PVT, asystole was significantly more common among non-survivors.

Children who undergo prolonged CPR generally have poorer outcomes compared to those who experience brief cardiac arrests. Research by Matos et al. [10] has demonstrated a concerning trend: the potential for survival decreases linearly by 2.1% for each minute of CPR administered within the critical window of 1 to 15 min. This finding underscores the importance of prompt and effective resuscitation, as the likelihood of favorable neurological outcomes also declines significantly, at a rate of 1.2% per minute. The study further revealed that the adjusted probability of survival is markedly affected by the duration of CPR. For instances where CPR was administered for durations ranging from 1 to 15 min, the probability of survival stood at 41%. However, this likelihood plummets to just 12% for CPR durations exceeding 35 min, highlighting a crucial threshold beyond which outcomes diminish substantially. In alignment with these findings, our own analysis indicated that the duration of CPR was significantly longer in non-survivors compared to survivors, with median durations of 45 min versus 15 min, respectively (*p* < 0.001).

The effect of serum lactate levels on clinical outcomes has been the subject of extensive research for many years. Initial studies revealed that elevated blood lactate levels could correlate with the duration of no-flow and low-flow states during CPR in animal models [11,12]. Weil et al. [13] reported that lactate levels measured within 10 min after the initiation of CPR might be significantly linked to survival outcomes in patients experiencing cardiac arrest. This early assessment of lactate levels can provide critical insights into the effectiveness of resuscitation efforts and the patient’s physiological status immediately following the event. Building on these foundational findings, subsequent studies conducted primarily in adult populations have further confirmed the relationship between lactate levels and clinical outcomes. Research has consistently shown that lower lactate levels post-CPR are associated with higher rates of survival as well as favorable neurological outcomes [14,15,16,17]. These studies emphasize the importance of monitoring lactate as a biomarker, as it reflects not only the metabolic state of the patient but also the adequacy of perfusion during and after resuscitation. However, studies examining this topic in pediatric populations are relatively limited. A notable single-center study conducted at a pediatric heart center demonstrated that higher serum lactate levels during cardiac arrest, along with increased doses of epinephrine administered, were significantly associated with increased mortality rates [18]. Another pediatric study [19] found that both the initial serum lactate level and the Glasgow Coma Scale (GCS) score prior to therapeutic hypothermia were significantly associated with favorable neurological outcomes at six months in pediatric patients who experienced out-of-hospital cardiac arrest and received therapeutic hypothermia. This highlights the importance of assessing these two metrics early in the resuscitation process, as they can provide valuable prognostic information regarding a child’s potential for recovery. Correspondingly, our results indicate that survivors had significantly lower lactate levels compared to non-survivors.

There is currently no established cutoff point for lactate levels in pediatric patients to effectively estimate survival probabilities. Previous studies conducted on adults have produced variable results regarding this issue. For example, Wang et al. [17] found that a lactate level of less than 9 mmol/L was significantly linked to higher survival probabilities; however, this threshold was higher than those reported in other studies. Seeger et al. [20] identified that a lactate level exceeding 6.94 mmol/L was associated with poor neurological outcomes, while Kaji et al. [21] reported that a lactate level below 5 mmol/L correlated with favorable neurological outcomes. The discrepancies in these threshold values may be attributed to differences in the timing of the lactate measurements and the criteria used for determining outcomes. In the present study, the median lactate level was 10.6 (3.2–23) mmol/L in the survival group. Research conducted in larger cohorts may be instrumental in establishing a cutoff value for pediatric populations.

### Limitations and Strengths

Our study has several limitations that should be carefully acknowledged in the context of our findings. First and foremost, the retrospective design of this study inherently restricts our ability to establish causal relationships between the observed variables. Additionally, the relatively small sample size of our cohort may limit the generalizability of our findings, which means that the causal–consecutive relationship analyzed in this study should be taken with caution due to the limited number of subjects. Moreover, it is important to highlight that data on pediatric cardiac arrest are notably limited in developing countries. Most large datasets and studies on this topic tend to originate from countries with advanced healthcare systems, such as the USA, where resources for research and patient care are more readily available. Consequently, the unique challenges and characteristics of pediatric cardiac arrest in developing regions may be underrepresented in the existing literature. In this regard, the contribution of our study is significant, as it provides valuable data from a developing country.

## 5. Conclusions

In the present study, the presence of asystole, high lactate levels, and prolonged CPR were identified as prognostic factors for in-hospital cardiac arrests (IHCAs). These findings may serve as valuable indicators of a patient’s condition and potential for recovery, enabling healthcare providers to make informed decisions regarding treatment strategies. By prioritizing these metrics, clinicians can better allocate resources and tailor interventions to meet the specific needs of pediatric patients experiencing cardiac arrest. In conclusion, IHCAs continue to represent a critical concern within pediatric healthcare, posing significant risks to patient outcomes. The variability in outcomes following IHCAs is influenced by a multitude of factors, including the underlying cause of the cardiac event, the initial cardiac rhythm observed, and the timeliness of the interventions provided.

## Figures and Tables

**Figure 1 medicina-60-01739-f001:**
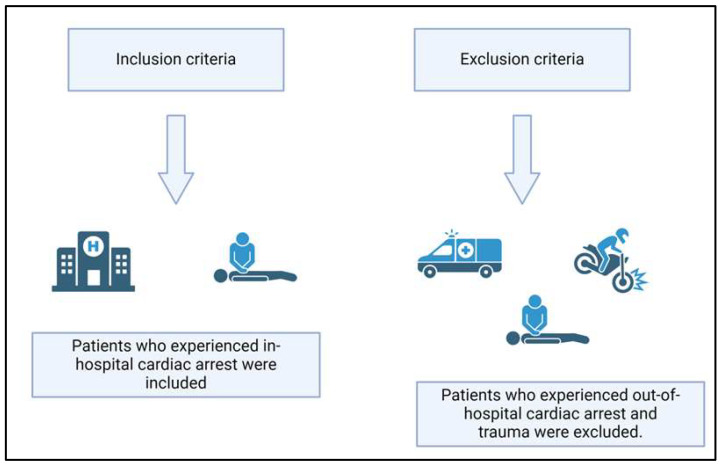
Schematic figure of inclusion and exclusion criteria.

**Table 1 medicina-60-01739-t001:** Comorbidities of patients with in-hospital cardiac arrest.

Diseases	*n* (%)
Neurologic disorders • Epilepsy • Cerebral palsy and epilepsy • Cerebral palsy • Brain tumor • Myopathy • Hypotonia • Spinal muscular atrophy type 1	15 (34.9) 2 6 3 1 1 1 1
Cardiac diseases • Arrythmia • Congenital heart disease	4 (9.3) 2 2
Metabolic disease (nonketotic hyperglycinemia)	1 (2.3)
Genetic disease (Down syndrome)	1 (2.3)
Renal disease (chronic renal failure)	1 (2.3)
Ocular disease (cataract)	1 (2.3)

**Table 2 medicina-60-01739-t002:** Comparison of parameters between survivors and non-survivors.

	All Patients (*n* = 43)	Survivors (*n* = 15)	Non-Survivors (*n* = 28)	*p* Values
Sex (Female/Male) *	22/21	7/8	15/13	0.66
Age, months **	36 (1–203)	36.5 (1–203)	16 (1–183)	0.98
Presence of comorbidities *, *n* (%)	23 (53.5)	10 (66.7)	13 (46.4)	0.20
Reason for cardiopulmonary resuscitation *, *n* (%)				
Pulselessness	23 (53.5)	6 (40)	17 (60.7)	0.08
Respiratory arrest	13 (30.2)	4 (26.7)	9 (32.1)	
Bradycardia	7 (16.3)	5 (33.3)	2 (7.1)	
The first monitored rhythm * (*n* (%)				
Asystole	29 (67.4)	6 (40)	23 (83.1)	0.005
Bradycardia	14 (32.6)	9 (60)	5 (17.9)	
Lactate levels (mmol/L) **	14.2 (1.2–27)	10.6 (3.2–23)	16.6 (1.2–27)	0.04
Duration of CPR **	45 (2–120)	15 (2–55)	45 (45–120)	<0.001

* These variables were compared using the Chi-square test. ** These variables were compared using the Mann–Whitney U test.

## Data Availability

Data are contained within the article.

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
