# Peer review of "Pediatric In-Hospital Cardiac Arrest: An Examination of Resuscitation Outcomes"

_medicina, 2024, doi:10.3390/medicina60111739_

Round 1
Reviewer 1 Report
Comments and Suggestions for Authors
The study "Pediatric In-Hospital Cardiac Arrest: An Examination of Resuscitation Outcomes" analyzes factors influencing survival rate in paediatric patients with IHCA. The authors used well-known approaches for selecting patients and data analysis. The results are concise and clear. Discussion is well written and conclusions are in accordance to the study findings.
There are, however, certain points requiring attention.
Line 98 - Please specify (and insert citation) which guidelines were used to make decisions of terminating resuscitation.
Lines 113 and 114 - Were both Kolmogorov-Smirnov and Shapiro-Wilk tests used in this study. In general, Shapiro-Wilk test is recommended for samples with 50 or less subjects in size.
Lines 118-120 - Why you did not add interquartile ranges. Those numbers could help visualize the dispersion of non-normally distributed data.
Lines 141 and 142 - Please consider writing numbers up to 10 in words.
The authors stated that in this study CPR was initiated due to pulselessness, respiratory arrest, and bradycardia. It is unclear, however, what guidelines were followed in order to determine the criteria for the initiation of CPR.
It is also needed for the authors to state what were the indications for hospital admission and if all the patients having IHCA during the five-and-a-half-year follow-up period were included in the analysis or some kind of sampling method was applied.
Due to the fact that the inclusion and exclusion criteria are very concise and clear, the schematic figure depicting inclusion is not necessary in this case. The authors may consider the possibility of removing Figure 1.
Also, if any of the patients previously had cardiac arrest, it should also be emphasized in the Results section.
It should be clearly stated that the causal-consecutive relationship analyzed in this study should be taken with caution due to the relatively small number of subjects.
Were there any parameters of neurological recovery also assessed? If yes, it would be beneficial if the authors could add some of those findings.
Other than these recommendations, further corrections are not necessary, and this study may be considered for publication.
Author Response
Comment 1: The study "Pediatric In-Hospital Cardiac Arrest: An Examination of Resuscitation Outcomes" analyzes factors influencing survival rates in paediatric patients with IHCA. The authors used well-known approaches for selecting patients and data analysis. The results are concise and clear. Discussion is well written, and conclusions are in accordance to the study findings.
Response 1: Thank you for your valuable comments.
Comment 2: There are, however, certain points requiring attention.
Response 2: I hope we have made the revisions in accordance with your suggestions.
Comment 3: Line 98 - Please specify (and insert citation) which guidelines were used to make decisions of terminating resuscitation.
Response 3: We used the following guidelines, and we cited them in the text (see references 5 and 6):
“Part 4: Pediatric Basic and Advanced Life Support: 2020 American Heart Association Guidelines for Cardiopulmonary Resuscitation and Emergency Cardiovascular Care” and “Part 6: Resuscitation Education Science: 2020 American Heart Association Guidelines for Cardiopulmonary Resuscitation and Emergency Cardiovascular Care”
Comment 4: Lines 113 and 114 - Were both Kolmogorov-Smirnov and Shapiro-Wilk tests used in this study. In general, Shapiro-Wilk test is recommended for samples with 50 or less subjects in size.
Response 4: You are right; we typically use the Kolmogorov-Smirnov and Shapiro-Wilk tests where appropriate, but in this study, we used the Shapiro-Wilk test. We clarified this issue as follows:
“In addition, analytical assessments were performed using the Shapiro-Wilk tests, which provided statistical measures to further validate whether the data followed a normal distribution pattern.”
Comment 5: Lines 118-120 - Why did you not add interquartile ranges. Those numbers could help visualize the dispersion of non-normally distributed data.
Response 5: You used minimum and maximum values to reflect the full range of the data. However, providing the interquartile range is another method. If desired, that can also be provided.
Comment 6: Lines 141 and 142 - Please consider writing numbers up to 10 in words.
Response 6: Thank you for your valuable comments. We wrote numbers up to 10 in words
Comment 7: The authors stated that in this study CPR was initiated due to pulselessness, respiratory arrest, and bradycardia. It is unclear, however, what guidelines were followed in order to determine the criteria for the initiation of CPR.
Response 7: We used the following guidelines, and we cited them in the text (see references 5 and 6):
“Part 4: Pediatric Basic and Advanced Life Support: 2020 American Heart Association Guidelines for Cardiopulmonary Resuscitation and Emergency Cardiovascular Care” and “Part 6: Resuscitation Education Science: 2020 American Heart Association Guidelines for Cardiopulmonary Resuscitation and Emergency Cardiovascular Care”
Comment 8: It is also needed for the authors to state what were the indications for hospital admission and if all the patients having IHCA during the five-and-a-half-year follow-up period were included in the analysis or some kind of sampling method was applied.
Response 8: We included all patients and clarified this issue as follows:
“This retrospective, single-center observational study was conducted to investigate all pediatric patients who experienced in-hospital cardiac arrests (IHCAs) over a five-and-a-half-year period, from January 2019 to July 2024.”
Comment 9: Due to the fact that the inclusion and exclusion criteria are very concise and clear, the schematic figure depicting inclusion is not necessary in this case. The authors may consider the possibility of removing Figure 1.
Response 9: Thank you for your valuable suggestion; however, we would prefer to retain the visual representation, as it facilitates a clearer understanding of the content.
Comment 10: Also, if any of the patients previously had cardiac arrest, it should also be emphasized in the Results section.
Response 10: None of the patients had previously experienced cardiac arrest. We added to the result section according to your comment (see line 148).
“None of the patients had previously experienced cardiac arrest.”
Comment 11: It should be clearly stated that the causal-consecutive relationship analyzed in this study should be taken with caution due to the relatively small number of subjects.
Response 11: We stated this issue in the limitation part as follows:
“………... Additionally, the relatively small sample size of our cohort may limit the generalizability of our findings, which means that the causal-consecutive relationship analyzed in this study should be taken with caution due to the limited number of subjects.”
Comment 12: Were there any parameters of neurological recovery also assessed? If yes, it would be beneficial if the authors could add some of those findings.
Response 12: This is a good point. We stated this issue in the result part as follows (see line 178):
“All survivors achieved a good neurological outcome upon discharge.”
Comment 13: Other than these recommendations, further corrections are not necessary, and this study may be considered for publication.
Response 13: Thank you for your valuable comments. I hope we have made the revisions in accordance with your suggestions.
Reviewer 2 Report
Comments and Suggestions for Authors
Dear authors,
I appreciate your efforts in creating this article. Any study that brings new information about cardiopulmonary resuscitation in children is always welcome in scientific publications.
The introduction effectively highlights the key elements of interest and the purpose of the study. Similarly, the materials and methods are correctly described, and they align with the type of study conducted. I particularly appreciated the diagram outlining the inclusion/exclusion criteria for the study.
However, I have noticed significant errors in the Results section.
In line 149, you mention that 23 patients had comorbidities, but in Table 1, it appears as if 42 patients are indicated. I believe this table should be revised and better structured for clarity.
Additionally, in Table 2, you state that 13 patients experienced respiratory arrest, but the columns show 5 survivors and 2 non-survivors. Furthermore, 7 patients are reported to have had bradycardia, yet the number of survivors is 4 and non-survivors is 9. This makes no sense.
Although you mentioned using statistical tests such as the Kolmogorov-Smirnov, Shapiro-Wilk, and Chi-square tests in the Methods section, they are not mentioned in the Results section.
The Discussion section needs revision as well, as there are some errors. For example, lines 198-205 repeat the same information twice. In line 211, the wording should be revised. Furthermore, when referencing other studies, it is better to present only the essence in percentages, without detailing all the numbers.
Author Response
Comment 1: I appreciate your efforts in creating this article. Any study that brings new information about cardiopulmonary resuscitation in children is always welcome in scientific publications.
Response 1: Thank you for your valuable comments.
Comment 2: The introduction effectively highlights the key elements of interest and the purpose of the study. Similarly, the materials and methods are correctly described, and they align with the type of study conducted. I particularly appreciated the diagram outlining the inclusion/exclusion criteria for the study.
Response 2: Thank you for your valuable comments.
Comment 3: However, I have noticed significant errors in the Results section. In line 149, you mention that 23 patients had comorbidities, but in Table 1, it appears as if 42 patients are indicated. I believe this table should be revised and better structured for clarity.
Response 3: Twenty-three patients were found to have comorbidities. The presentation of these details under the main headings may have contributed to some confusion. I trust that the revised version will offer greater clarity. Thank you.
Comment 4: Additionally, in Table 2, you state that 13 patients experienced respiratory arrest, but the columns show 5 survivors and 2 non-survivors. Furthermore, 7 patients are reported to have had bradycardia, yet the number of survivors is 4 and non-survivors is 9. This makes no sense.
Response 4: You are correct, and I appreciate your attention to detail. The numbers in the respiratory arrest and bradycardia columns were mistakenly reversed, but they have now been corrected. We apologize.
Comment 5: Although you mentioned using statistical tests such as the Kolmogorov-Smirnov, Shapiro-Wilk, and Chi-square tests in the Methods section, they are not mentioned in the Results section.
Response 5: We clarified this issue under table 2.
Comment 6: The Discussion section needs revision as well, as there are some errors. For example, lines 198-205 repeat the same information twice. In line 211, the wording should be revised. Furthermore, when referencing other studies, it is better to present only the essence in percentages, without detailing all the numbers.
Response 6: We removed the repeated parts. We presented only the essence in percentages, without detailing all the numbers according to your comments.
Round 2
Reviewer 2 Report
Comments and Suggestions for Authors
Thank you for the revisions made to the text, in accordance with the recommendations. I hope the article receives many citations in the medical literature.
Author Response
Comment 1: Thank you for submitting this interesting study to the present special issue and for addressing the comments from reviewers. Please find some additional comments below that I believe will increase the clarity of the manuscript.
Response 1: Thank you for your valuable comments. I hope we have made the revisions in accordance with your suggestions.
Comment 2: Lines 162-163: You mention that "Upon arrival at the emergency department, the first monitored cardiac rhythm revealed that asystole was present in 29 patients, representing 67.4% of the cohort.". This suggests that patients arrived in cardiac arrest. However, in your exclusion criteria you mention that children with cardiac arrest prior to their arrival at the hospital were not included. Please rephrase lines 162-163 to better highlight that the cardiac arrest event happened in the emergency department.
Response 2: You are right; we meant the cardiac rhythm detected before CPR. We revised this issue according to your comment as follows:
“The first monitored cardiac rhythm detected before CPR revealed that asystole was present in 29 patients…..”
Comment 3: Regarding good neurological outcome, it would make sense to provide the scale used in Methods and provide numerical values in the Results section. When was neurological function assessed (time-wise)?
Response 3: We used Pediatric Cerebral Performance Category (PCPC) and we depicted this issue in the method section and results as follows:
“Neurological assessment was performed before discharge from the hospital, and the Pediatric Cerebral Performance Category (PCPC) was used to evaluate neurological outcomes.”
“Among the surviving patients, ten had a pre-existing chronic condition that affected their neurological development. No worsening of neurological findings was observed in these patients. The remaining five patients had normal neurological function as assessed by the PCPC.”
Comment 4: As you do not provide a follow-up on these patients, would it make more sense to name the groups "ROSC not achieved" and "ROSC achieved" instead of survivors and non-survivors? The lack of follow-up should also be mentioned as a limitation. If you indeed had a follow-up, then you should also provide the ROSC percentages. Also, this would provide better context when you mention that CPR time was greater in the non-survivor group. If this was the non-ROSC group, this is expected and is not useful. If however this was actually the non-survivor group, this finding is important to inform timeline cut-offs for CPR efforts.
Response 4: The patients were monitored from their admission to the emergency department until discharge. All of the survivors are indeed true survivors.
Comment 5: You mention sinus rhythm as an initial cardiac arrest rhythm. I believe it would be better to clarify that sinus rhythm was present in the context of pulseless electrical activity or bradycardia. Please clarify this.
Response 5: You are right, we corrected as bradycardia.
Comment 6: I believe that the conclusion section should be revised to better reflect the major findings of your study, their importance and possible future perspectives.
Response 6: Thank you for your valuable comments. I hope we have made the revisions in accordance with your suggestions. We revised as follows:
“In the present study, the presence of asystole, high lactate levels, and prolonged CPR were identified as prognostic factors for in-hospital cardiac arrests (IHCAs). These findings may serve as valuable indicators of a patient’s condition and potential for recovery, enabling healthcare providers to make informed decisions regarding treatment strategies. By prioritizing these metrics, clinicians can better allocate resources and tailor interventions to meet the specific needs of pediatric patients experiencing cardiac arrest. In conclusion, IHCAs continue to represent a critical concern within pediatric healthcare, posing significant risks to patient outcomes. The variability in outcomes following IHCAs is influenced by a multitude of factors, including the underlying cause of the cardiac event, the initial cardiac rhythm observed, and the timeliness of the interventions provided.”